# Reinvestigating the Parabolic-Shaped Eddy Viscosity Profile for Free Surface Flows

**Rafik Absi** 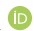

ECAM-EPMI, LR2E-Lab, Quartz-Lab (EA 7393), 13 Boulevard de l'Hautil, 95092 Cergy-Pontoise, France; rafik.absi@yahoo.fr or r.absi@ecam-epmi.com; Tel.: +33-1-30756921

**Abstract:** The flow in rivers is turbulent. The main parameter related to turbulence in rivers is the eddy viscosity, which is used to model a turbulent flow and is involved in the determination of both velocities and sediment concentrations. A well-known and largely used vertical distribution of eddy viscosity in free surface flows (open channels and rivers) is given by the parabolic profile that is based on the logarithmic velocity profile assumption and is valid therefore only in the log-law layer. It was improved thanks to the log-wake law velocity profile. These two eddy viscosities are obtained from velocity profiles, and the main shortcoming of the log-wake profile is the empirical Coles' parameter. A more rigorous and reliable analytical eddy viscosity model is needed. In this study, we present two analytical eddy viscosity models based on the concepts of velocity and length scales, which are related to the exponentially decreasing turbulent kinetic energy (TKE) function and mixing length, namely, (1) the exponential-type profile of eddy viscosity and (2) an eddy viscosity based on an extension of von Karman's similarity hypothesis. The eddy viscosity from the second model is $\mathrm{Re}_*$-independent, while the eddy viscosity from the first model is $\mathrm{Re}_*$-dependent (where $\mathrm{Re}_*$ is the friction Reynolds number). The proposed analytical models were validated through computation of velocity profiles, obtained from the resolution of the momentum equation and comparisons to experimental data. With an additional correction function related to the damping effect of turbulence near the free surface, both models are similar to the log-wake-modified eddy viscosity profile but with different values of the Coles' parameter, i.e., $\Pi = 0.2$ for the first model and $\Pi = 0.15$ for the second model. These values are similar to those found in open-channel flow experiments. This provides an explanation about the accuracy of these two analytical models in the outer part of free surface flows. For large values of $\mathrm{Re}_*$ ($\mathrm{Re}_* > 2000$), the first model becomes $\mathrm{Re}_*$ independent, and the two coefficients reach asymptotic values. Finally, the two proposed eddy viscosity models are validated by experimental data of eddy viscosity.

**Keywords:** river flow; open channels; eddy viscosity; parabolic profile; streamwise velocity distribution; turbulent kinetic energy (TKE); mixing length; log law; log wake

## 1. Introduction

Determination of velocity distribution in open-channel flows and rivers is a topic of high interest and is involved in different practical applications [1–7]. However, the hydrodynamic in rivers and open-channel flows is strongly influenced by turbulence [8–11]. Different experimental studies were conducted to better understand the effect of turbulence on the streamwise velocity distribution [12–16].

Open-channel flow studies are considered an important preliminary step to investigate more complex river flows. The main interest in laboratory investigations of free surface turbulent flows is related to the experimental conditions that are chosen to be in agreement with the assumptions related to the models. Therefore, experiments in laboratory flumes allowed analytical models of turbulence to be developed that are in the form of analytical solutions, semi-theoretical or empirical relationships [17–20]. These models were proposed for mean velocities, turbulent kinetic energy (TKE), mixing length, eddy viscosity with their

link to flow parameters (flow depth, friction velocity) [11,18]. The developed analytical models are mostly two-dimensional and for uniform flows, while free surface turbulent flows in rivers are generally strongly three-dimensional and non-uniform. Even if the assumptions are far from the real-life conditions in rivers, and the developed analytical models cannot account for the full complexity of turbulent river flows, these models present high practical interest. For example, measured data in the central part of rivers are well described by the analytical expressions obtained from laboratory open-channel flows [21].

In open-channel flows and rivers, in the classical two-layer approach, the flow is divided into two regions: an inner ($\xi < 0.2$) and an outer ($\xi > 0.2$) region (where $\xi = y/h$ is the ratio of the distance from the bed $y$ to flow depth $h$). For smooth open-channel flows, log law allows accurate description of mean streamwise velocities $U^+(y^+)$ in the logarithmic layer ($30 < y^+ < 0.2Re_*$ or $30/Re_* < \xi < 0.2$, where in wall unit $y^+ = y\,u_*/\nu$, $u_*$ is the friction velocity, $\nu$ the kinetic viscosity, and $Re_* = h\,u_*/\nu$ is the friction Reynolds number) [11]. For river flows, Franca and Lemmin [22] found from analysis of experimental data from a field study of extremely rough, three-dimensional river flows, that in more than 65% of the profiles, the log law can be applied up to $\xi = 0.4$, while above this value, mean velocities *show* deviations from the logarithmic profile.

In the outer region, in addition to the simple power law [23,24], the log-wake law is largely used. It is an extension of the log law by adding the Coles' wake function, which contains the Coles' parameter $\Pi$ [25,26]. However, this method is empirical and no physical-based approach is available to determine the Coles' parameter $\Pi$ [27]. The value of $\Pi$ is therefore not universal. From experiments in zero-pressure-gradient boundary layers, Cebeci and Smith [28] found that $\Pi$ increases with the Reynolds number and becomes $\Pi = 0.55$ at high Reynolds numbers. For open-channel flows over smooth beds, the following values were found: $\Pi$ increases from zero with $Re_*$ and becomes $\Pi \approx 0.2$ for $Re_* = 2000$ [18], $\Pi \approx 0.08$ [29], $\Pi = 0.1$ [30], $\Pi = 0.3$ [31], $\Pi = 0.45$ for velocity distribution with dip-phenomenon in narrow open channels [32].

However, the more rigorous method for the prediction of velocity profiles is based on the resolution of the momentum equation. This method needs a reliable model for the eddy viscosity, which is the main parameter related to turbulence used in free surface flows since it is involved in the determination of both velocities and sediment concentrations (through the sediment diffusivity, i.e., the product of the eddy viscosity by the inverse of the turbulent Schmidt number) [33,34]. A well-known and largely used vertical distribution of eddy viscosity in free surface flows (open channels and rivers) is given by the parabolic profile [11]. This profile is based on the logarithmic velocity profile assumption and is valid therefore only in the log-law layer. It was improved thanks to the log-wake law velocity profile. These two eddy viscosities are obtained from velocity profiles. As for velocity profiles in the outer region, the main shortcoming of the log-wake eddy viscosity profile is the non-universal Coles' parameter $\Pi$.

An analytical eddy viscosity model is therefore needed to predict velocity profiles. In this study, we present two eddy viscosity models based on the concepts of velocity and length scales, which are related, respectively, to the exponentially decreasing turbulent kinetic energy (TKE) function [11] and mixing length, namely, (1) the exponential-type profile of eddy viscosity [35,36] and (2) an eddy viscosity based on an extension of von Karman's similarity hypothesis [37–39]. An additional correction is used in order to account for the damping effect of turbulence near the free surface. The proposed analytical models are validated through computation of velocity profiles, obtained from the resolution of the momentum equation, and comparisons to experimental data. This study aims to provide an explanation and a theoretical foundation to the empirical well-known eddy viscosity profiles.

## 2. Literature for Eddy Viscosity Models for Open-Channel Flows

### 2.1. Parabolic Eddy Viscosity

The widely used eddy viscosity ($\nu_t$) formulation is the parabolic profile given by [40,41]

$$\nu_t(y) = \kappa u_* y \left(1 - \frac{y}{h}\right), \tag{1}$$

where $y$ is the vertical distance from the bed, $\kappa$ is the von Karman constant, $h$ the flow depth, and $u_*$ the friction or shear velocity. Equation (1) is based on shear stress, which decreases linearly with distance from channel bed $y$ and a logarithmic velocity profile [33].

### 2.2. Log-Wake-Modified Eddy Viscosity Profile

When used for the resolution of the momentum equation, the parabolic eddy viscosity profile (1) is unable to predict accurately velocities outside the log layer [36]. In order to improve the parabolic eddy viscosity (Equation (1)), it is corrected in accordance with Coles' log-wake law for velocities as [18]

$$\nu_t(y) = \frac{\kappa u_* y \left(1 - \frac{y}{h}\right)}{1 + \pi \, \Pi \, \frac{y}{h} \sin\left(\frac{\pi y}{h}\right)} \tag{2}$$

where $\Pi$ is the Coles' parameter. In Equation (2), the eddy viscosity (1) is corrected by dividing the parabolic eddy viscosity profile by the term $1 + \pi\Pi(y/h)\sin(\pi y/h)$ of the log-wake velocity profile.

### 2.3. Mixing Length and Mixing Velocity

In order to predict the velocity profile over the entire flow depth, it is more suitable to define the eddy viscosity from the concepts of velocity and length scales, which are here given by mixing velocity and mixing length as

$$\nu_t = w_m l_m \tag{3}$$

From the parabolic profile given by Equation (1), mixing length and mixing velocity are given, respectively, by $l_m = \kappa y \sqrt{1 - y/h}$ and $w_m = u_* \sqrt{1 - y/h}$. The eddy viscosity of Equation (2) allows for the following expression:

$$l_m(y) = \frac{\kappa y \sqrt{1 - y/h}}{1 + \pi\Pi(y/h)\sin(\pi y/h)} \tag{4}$$

Equation (4) consists of a correction of the "parabolic" mixing length $\kappa y \sqrt{1 - y/h}$ by dividing it by the term $1 + \pi\Pi(y/h)\sin(\pi y/h)$ as in Equation (2). The related mixing velocity is given by $w_m = u_* \sqrt{1 - y/h}$ [42].

## 3. Proposed Eddy Viscosity Models for Free Surface Flows

The eddy viscosity is related to turbulent kinetic energy (TKE) as

$$\nu_t = C_\mu^{\frac{1}{4}} \sqrt{k} l_m \tag{5}$$

where $k$ is the turbulent kinetic energy (TKE), and $C_\mu = 0.09$.

### 3.1. Mixing Velocity from TKE Profile

A semi-theoretical function for TKE is given by [11]:

$$k(\xi) = D_k u_*^2 e^{-2C_k \xi} \tag{6}$$

where $\xi = y/h$, $C_k$ and $D_k$ are empirical constants, $D_k = 4.78$, and $C_k = 1$ [11]. Equation (6) was validated by direct numerical simulation DNS data [43]. Instead of

the mixing velocity given in the above section by $w_m = u_* \sqrt{1 - y/h}$, the shape of mixing velocity should be supported by turbulence intensity measurements [38] and in agreement with the TKE formulation of Equation (6).

From Equations (3) and (5), the mixing velocity is related to TKE as follows:

$$w_m = C_\mu^{\frac{1}{4}} \sqrt{k} = C_\mu^{\frac{1}{4}} \sqrt{D_k} u_* e^{-C_k \xi} \qquad (7)$$

With the assumption $\sqrt{D_k} = 1/C_\mu^{\frac{1}{4}}$ (based on log-law and local equilibrium assumption [39]) and with $C_k = 1$, the mixing velocity reverts to

$$w_m(\xi) = u_* e^{-\xi}, \qquad (8)$$

which shows that the mixing velocity decreases exponentially with distance from the bed and is in agreement with observations of turbulence intensity and TKE.

### 3.2. Damping Function for Free Surface

An additional correction is required in order to account for the damping effect of turbulence near the free surface [44–46]. In order to decrease turbulent viscosity near the free surface, Hosoda [47] proposed a damping function as

$$f(\xi) = 1 - e^{-B_f(1-\xi)} \qquad (9)$$

where $B_f$ is a damping coefficient.

If we include the free surface damping function in Equation (3), the eddy viscosity reverts to

$$\nu_t = w_m l_m f(\xi) \qquad (10)$$

In the following sections, two eddy viscosity formulations will be presented.

### 3.3. First Formulation: Exponential-Type Profile of Eddy Viscosity

In the equilibrium region, where TKE production is balanced by dissipation, the velocity gradient is given by $\frac{dU}{dy} = \frac{C_\mu^{1/4} \sqrt{k}}{l_m}$. In the log-law layer, $\frac{dU}{dy} = \frac{u_*}{\kappa y}$, and with a TKE given by $\sqrt{k} = C_\mu^{-1/4} u_* f(y)$, the mixing length should read as $l_m = \kappa y f(y)$ [38]. Since in the equilibrium region ($y^+ > 50$), TKE is given by Equation (6), and the velocity profile is given by the log law for ($y^+ > 30$). These two conditions that are given by $\frac{dU}{dy} \approx \frac{\sqrt{k}}{l_m} = \frac{u_*}{\kappa y}$ show that the mixing length should be as $l_m = \kappa y e^{-C_k(y/h)}$. The 1st eddy viscosity is therefore given by

$$\nu_t(y) = \alpha_1 \kappa u_* y e^{-C_1 \xi} \qquad (11)$$

where $\alpha_1$ and $C_1$ are two coefficients, $\alpha_1$ is related to $C_\mu$ and $D_k$, while $C_1$ is related to $C_k$. Equation (11), i.e., the exponential-type profile of eddy viscosity is consistent with the exponentially decreasing mixing velocity (8). Equation (11) was proposed empirically and was used in the planetary boundary layer [48] and coastal engineering [49–53]. However, in order to allow accurate description for different flow conditions, Equation (11) was written in a $Re_*$-dependent form as [35,36]

$$\nu_t(y) = u_* y e^{-\frac{y^+ + 0.34 Re_* - 11.5}{0.46 Re_* - 5.98}} \qquad (12)$$

where in the wall units, $y^+ = y u_* / \nu$, and the friction Reynolds number $Re_* = h u_* / \nu$. In other words,

$$\nu_t^+ (y^+) = y^+ e^{-\frac{y^+ + 0.34 Re_* - 11.5}{0.46 Re_* - 5.98}}$$

where $\nu_t{}^+ = \nu_t/\nu$. The link between Equations (11) and (12) is given by

$$C_\alpha = \alpha_1 \kappa = e^{-\frac{0.34Re_* - 11.5}{0.46Re_* - 5.98}} \text{ and } C_1 = \frac{Re_*}{0.46Re_* - 5.98}$$

*3.4. Second Formulation: Eddy Viscosity Formulation Based on Miwing Length Equation from Similarity Hypothesis*

This formulation uses a mixing length, which was derived from an extension of von Karman's similarity hypothesis, energy equilibrium assumption, and Nezu and Nakagawa's (1993) TKE function as [37–39,54] (Appendix A)

$$l_m(\xi) = \kappa h \left(1 - e^{-\xi}\right) \tag{13}$$

Using Equations (8)–(10) and (13), the second proposed eddy viscosity is given by

$$\nu_t(\xi) = \kappa h u_* e^{-\xi} \left(1 - e^{-\xi}\right) \left(1 - e^{-B_f(1-\xi)}\right) \tag{14}$$

## 4. Results

The following ordinary differential equation for velocity distribution $U$ in open-channel flows was obtained from analysis of the Reynolds-averaged Navier–Stokes equations [32,55].

$$\frac{dU}{dy} = \frac{u_\tau^2}{\nu + \nu_t} \left[ \left(1 - \frac{y}{h}\right) - \alpha \frac{y}{h} \right] \tag{15}$$

where $\alpha$ is a parameter related to dip-phenomenon. For wide-open channels (ratio of channel width to flow depth >5) $\alpha = 0$, and Equation (15) reverts in the outer region (in the wall units) to the following:

$$\frac{dU^+}{dy^+} = \frac{1}{\nu_t{}^+} \left(1 - \frac{y^+}{Re_*}\right) \tag{16}$$

Mean streamwise velocities are obtained from the numerical resolution of Equation (16). To solve Equation (16), the eddy viscosity $\nu_t{}^+$ is calculated using the two proposed analytical eddy viscosity models given above, i.e., the first is given by Equation (12), while the second by Equation (14).

Both proposed models are validated by experimental data of velocities in open-channel flows for $923 < Re_* < 6139$ [18]. The measurements were carried out in a rectangular cross section and a hydraulically smooth wall open channel. The total length of the channel is 20 m with a cross-sectional size (60 cm wide × 65 cm deep). The width of the channel is sufficiently large to neglect the wall effect. A flow depth of 10 cm was kept constant with varying discharge to examine at various Froude and Reynold numbers. Moreover, measurement was carried out at a length of 18 m from the inlet position. Laser Doppler Anemometer (LDA) was used to carry out velocity measurements under different flow conditions. Table 1 summarizes experimental hydraulic conditions. The flow conditions are listed in Table 1 [18].

**Table 1.** Flow conditions [18].

| Case | Depth of Flow, h (cm) | Width to Depth Ratio | Reynolds Number [1], $Re = \frac{4RU_m}{\nu}$ | Froude Number, $Fr = \frac{U_m}{\sqrt{gh}}$ | Friction Reynolds Number, $Re_* = \frac{hu_*}{\nu}$ |
|------|------|------|------|------|------|
| P2 | 10.3 | 5.9 | $5.5 \times 10^4$ | 0.189 | 923 |
| P3 | 10.0 | 6.0 | $14.3 \times 10^4$ | 0.488 | 2156 |
| P4 | 10.0 | 6.0 | $21.0 \times 10^4$ | 0.704 | 3001 |
| P5 | 10.5 | 5.7 | $44.0 \times 10^4$ | 1.170 | 6139 |

[1] $U_m$ = mean bulk velocity, $R$ = hydraulic radius.

### 4.1. Velocity Profiles from the First Eddy Viscosity Formulation: Exponential-Type Profile

Computed mean streamwise velocity profiles are obtained from (16) with the first eddy viscosity given by Equation (12) and are validated by experimental data [18]. The following boundary condition for the velocity is applied at $\xi = 0.2$ (or $y^+ = 0.2\,Re_*$): $U^+(y^+ = 0.2Re_*) = (1/\kappa)\ln(0.2Re_*) + B$; where $\kappa = 0.41$ and $B = 5.29$ [11].

Figures 1 and 2 show comparisons between computed velocity profiles (red solid lines) and experimental data (symbols). The model allows the prediction of log law (black thin dashed lines) to be improved in the outer region.

In order to improve the results, the first eddy viscosity given by (12) is used with the condition of an eddy viscosity equal to zero at the free surface, which requires the use of the damping function given by Equation (9), Equation (12) reverts to

$$\nu_t^{\,+} = y^+ e^{-\frac{y^+ + 0.34Re_* - 11.5}{0.46Re_* - 5.98}}\left(1 - e^{-B_f\left(1 - \frac{y^+}{Re_*}\right)}\right) \tag{17}$$

Results obtained with Equation (17) (magenta thick dashed curves) allow the prediction to be improved, particularly for high Reynolds numbers.

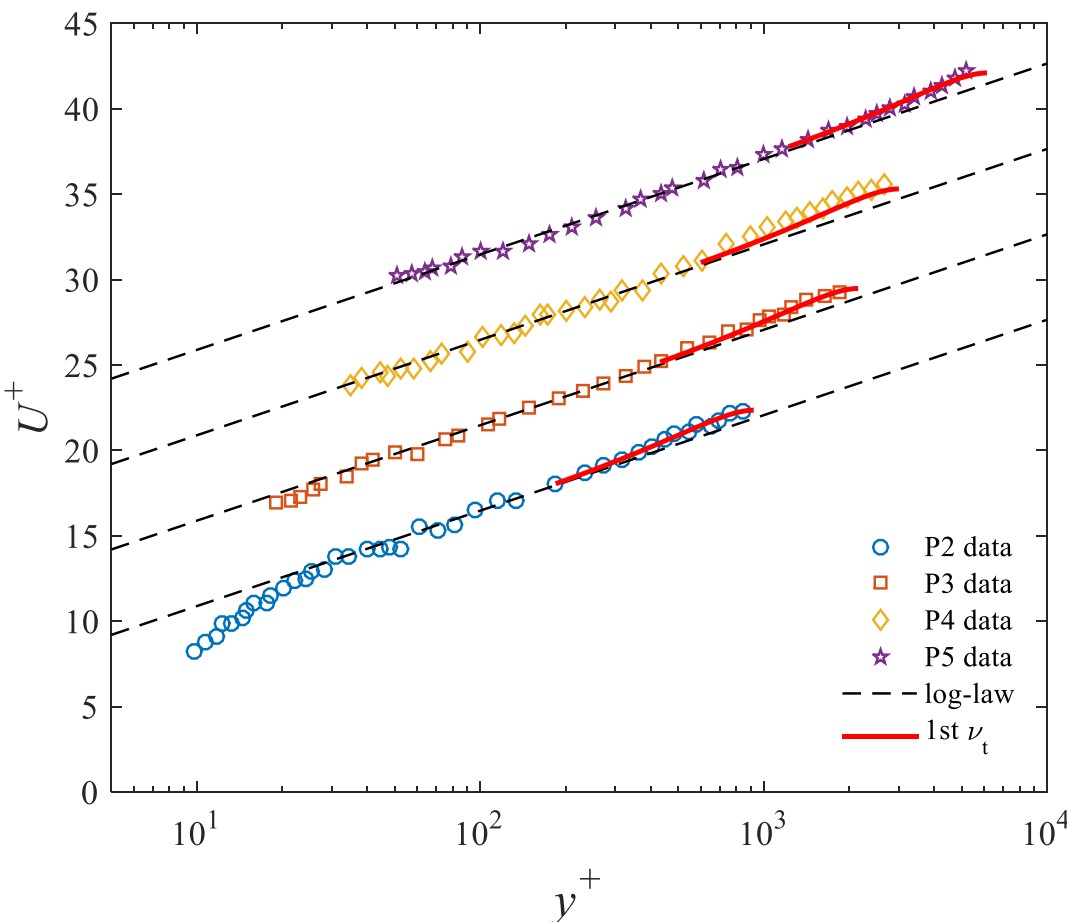

**Figure 1.** Velocity profiles (red solid lines) obtained from (16) with the first eddy viscosity (12); symbols: experimental data [18]; dashed lines: log law (profiles shifted by 5 units).

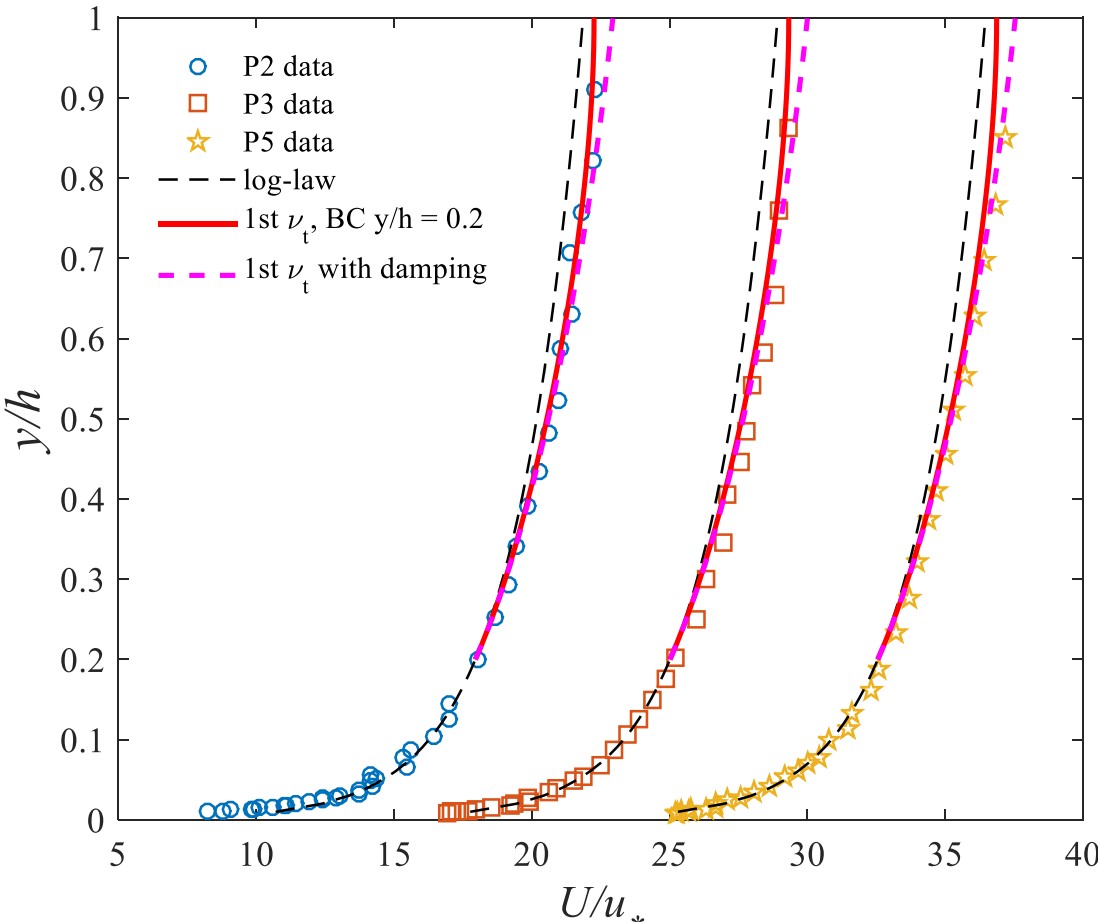

**Figure 2.** Velocity profiles obtained from (16) with the first eddy viscosity (12) (red solid lines) and from (16) with (17) ($B_f = 6$) magenta thick dashed lines; symbols: experimental data [18]; dashed lines: log law (profiles shifted by 5 units).

### 4.2. Velocity Profiles from the Second Eddy Viscosity Formulation Based on Von Karman's Similarity Hypothesis

Computed mean streamwise velocity profiles are obtained from (16), and the second eddy viscosity given by (14) and are validated by experimental data [18] (Table 1). The same boundary condition is applied (velocity equal to the log-law value at $y^+ = 0.2\, Re_*$). Figure 3 shows comparisons between computed velocity profiles (red solid lines) and experimental data (symbols). The model allows log-law profiles (black thin dashed lines) to be improved in the outer region. Figure 3 shows that computed velocity profiles (red solid lines) show good agreement and improve the prediction of log law (black dashed thin lines). In order to improve the results, the momentum equation is resolved with a second boundary condition at the lower limit of the logarithmic layer, i.e., $y^+ = 30$, where the velocity is given by the log law as $U^+(30) = (1/\kappa)\ln(30) + B$. Results (magenta thick dashed curves) allow the prediction to be improved, particularly for high Reynolds numbers.

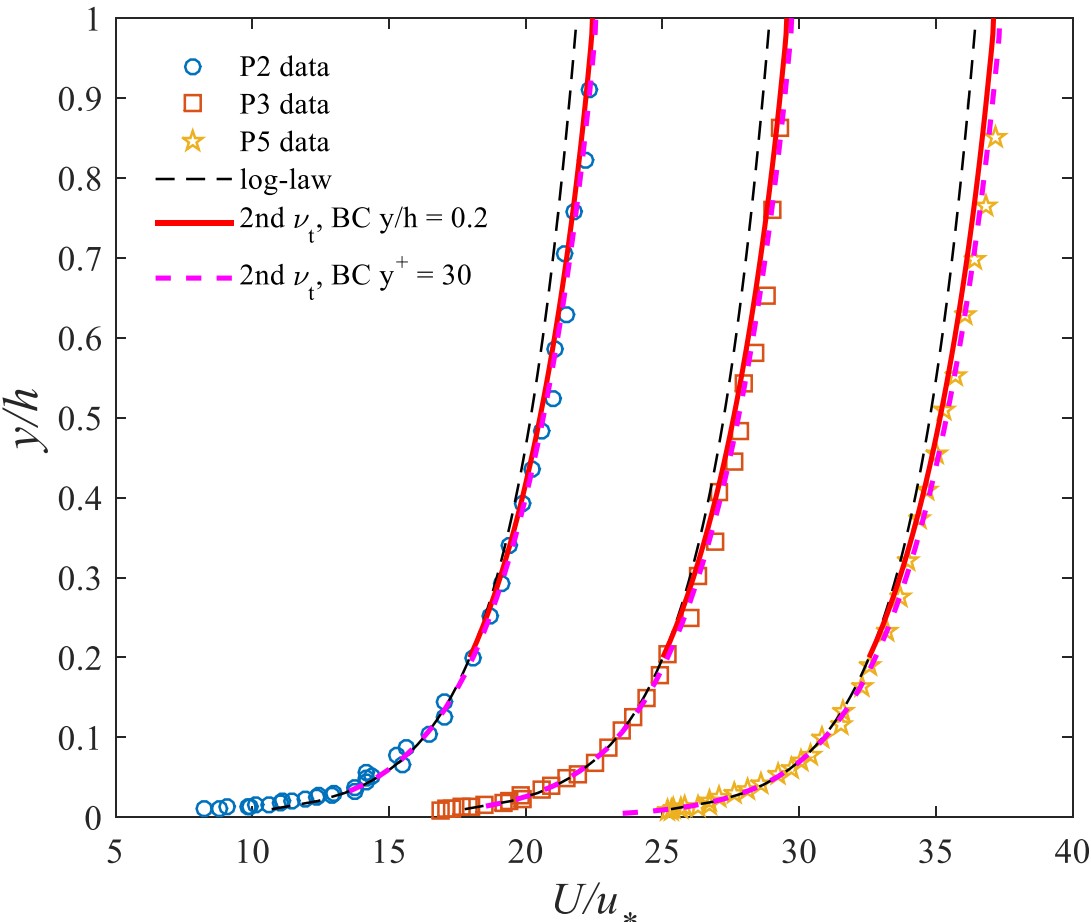

**Figure 3.** Velocity profiles obtained from (16) with the second eddy viscosity (14) ($B_f = 4$) with the first boundary condition at $\xi = 0.2$ (red solid lines), and the second boundary condition at $y^+ = 30$ (magenta thick dashed lines); symbols: experimental data [18]; dashed lines: log law (profiles shifted by 5 units).

### 4.3. Eddy Viscosity Profiles

In order to compare the two proposed analytical eddy viscosity models (Equations (12), (14) and (17)) with the existing parabolic and log-wake-modified eddy viscosity profiles (Equations (1) and (2)), all equations are written in the same dimensionless form as

$$\frac{\nu_t}{h\,u_*} = \kappa \frac{y}{h}\left(1 - \frac{y}{h}\right) \tag{18}$$

$$\frac{\nu_t}{h\,u_*} = \frac{\kappa \frac{y}{h}\left(1 - \frac{y}{h}\right)}{1 + \pi\Pi\frac{y}{h}\sin\left(\frac{\pi y}{h}\right)} \tag{19}$$

$$\frac{\nu_t}{h\,u_*} = \xi e^{-\frac{(\xi+0.34)Re_*-11.5}{0.46Re_*-5.98}} \tag{20}$$

$$\frac{\nu_t}{h\,u_*} = \xi e^{-\frac{(\xi+0.34)Re_*-11.5}{0.46Re_*-5.98}}\left(1 - e^{-B_f(1-\xi)}\right) \tag{21}$$

$$\frac{\nu_t}{h\,u_*} = \kappa e^{-\xi}\left(1 - e^{-\xi}\right)\left(1 - e^{-B_f(1-\xi)}\right) \tag{22}$$

We notice that eddy viscosities from Equations (18), (19), and (22) are $Re_*$-independent, while Equations (20) and (21) are $Re_*$ dependent.

Figures 4 and 5 show comparisons of the vertical distribution of the different eddy viscosity models. The three $Re_*$-independent Equations (18), (19), and (22) are first compared.

Figure 4 shows that the shape of the eddy viscosity given by the second model (Equation (22)) (red solid line) is similar to the parabolic profile (green dashed line), where the maximum value is located at the same position, i.e., half water depth ($\xi = 0.5$). Even though the second model exhibits a similar shape, it predicts smaller values than the parabolic profile. The profile obtained from the second model is compared to log-wake-modified (dash-dotted lines) profiles. The magenta dash-dotted curve is from the log-wake-modified eddy viscosity given by Equation (19) with a Coles' parameter $\Pi = 0.2$. With a smaller value of $\Pi$, the blue dash-dotted curve (for $\Pi = 0.15$) is closer to the eddy viscosity given by the second model. This value ($\Pi = 0.15$) is close to values found for open-channel flow experiments [18,30]. This provides an explanation about the accuracy of the computed velocity profiles obtained by the second model in the outer part of free surface flows.

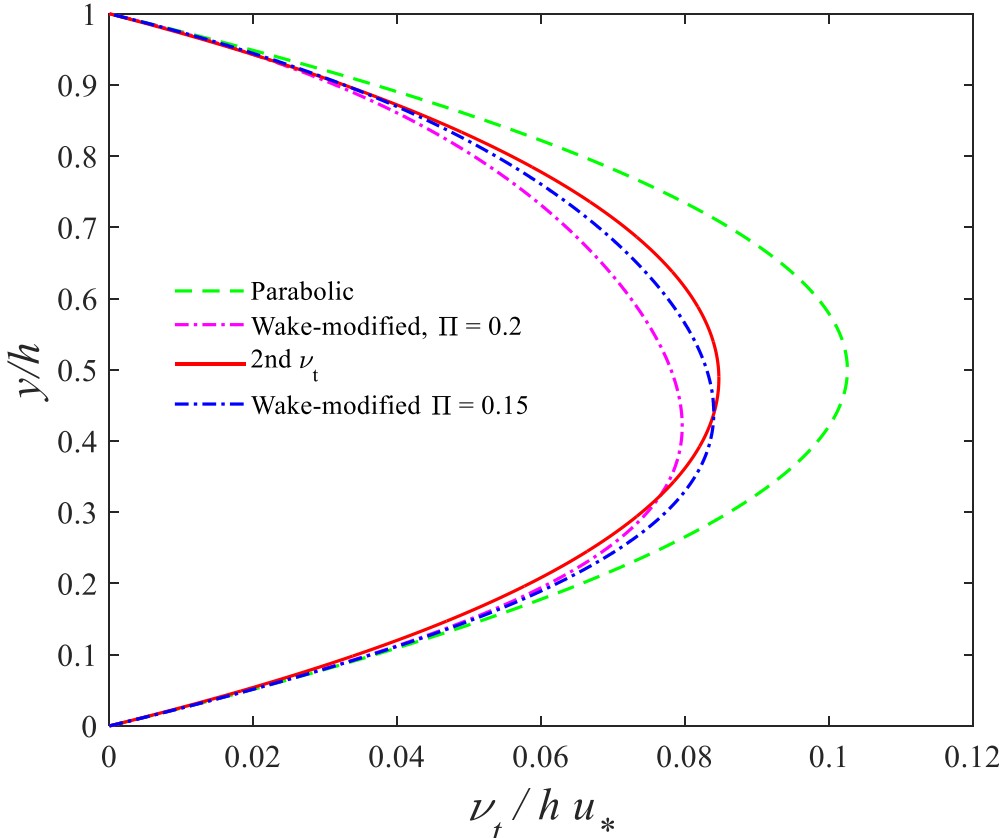

**Figure 4.** Vertical distribution of eddy viscosity; green dashed line: parabolic eddy viscosity (Equation (18)); dash-dotted lines: log-wake-modified eddy viscosity (Equation (19)); solid line: second eddy viscosity (Equation (22)) with $B_f = 4$.

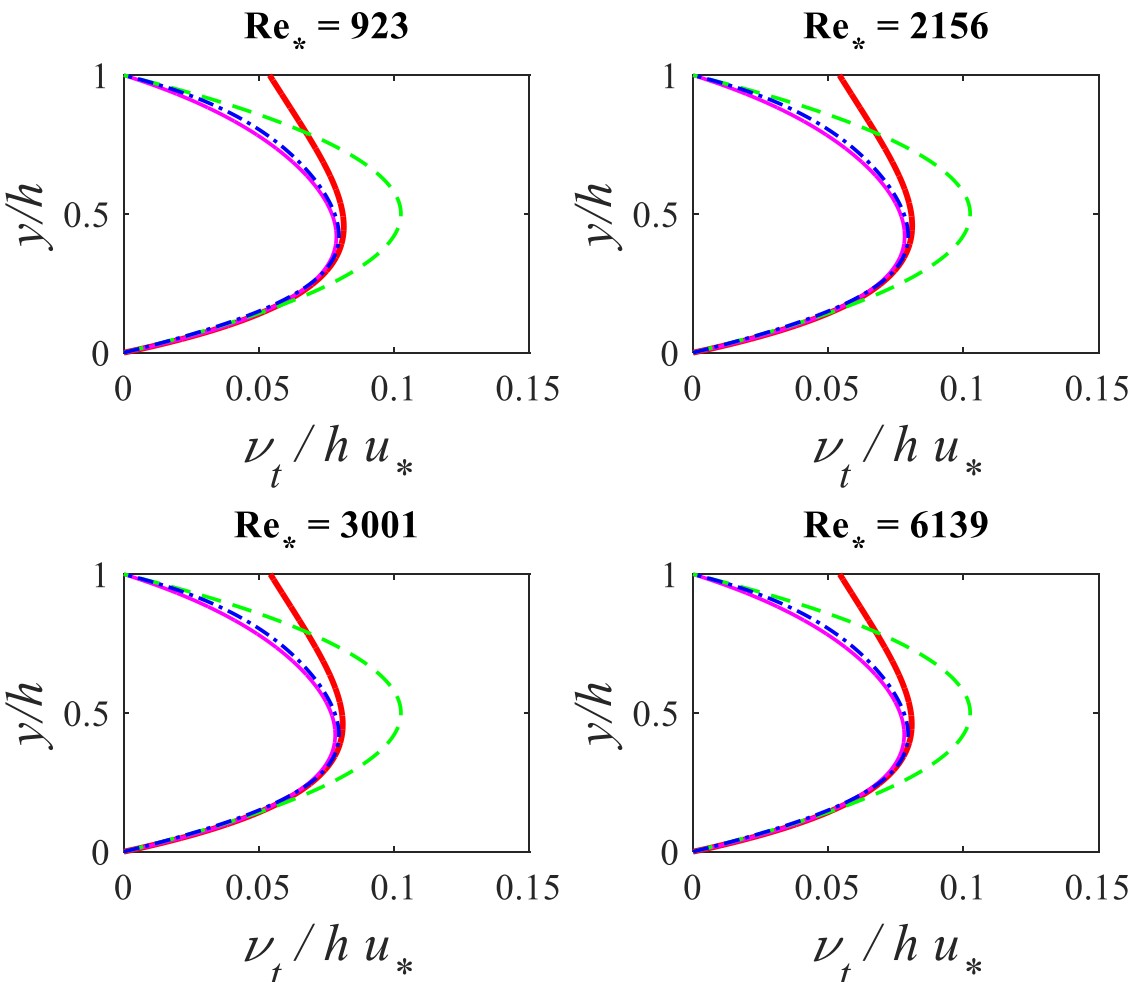

**Figure 5.** Eddy viscosity profiles; thick red solid lines: first eddy viscosity (Equation (20)); thin magenta solid lines: first eddy viscosity with free surface damping function (Equation (21)); blue dash-dotted lines: log-wake-modified eddy viscosity (Equation (19)) with $\Pi = 0.2$; green dashed lines: parabolic eddy viscosity (Equation (18)).

For the $Re_*$-dependent eddy viscosity given by the first model (Equations (20) and (21)), Figure 5 shows comparisons for the vertical distribution of the eddy viscosity for the four friction Reynolds number. The eddy viscosity curves (thick red solid lines) obtained from the first model (Equation (20)) show that the eddy viscosity increases from the bed—the maximum value is located at around the half water depth and then decreases until the free surface. However, the eddy viscosity does not vanish at the free surface. With the condition of an eddy viscosity equal to zero at the free surface, the profiles with the damping function (Equation (21)) predict everywhere smaller values than the parabolic eddy viscosity (thin magenta solid lines) and have a shape similar to the log-wake-modified eddy viscosity profile (blue dash-dotted lines). Interestingly, with a value of Coles' parameter $\Pi = 0.2$, eddy viscosities from both the first model (Equation (21)) and log-wake-modified profile (Equation (19)) are almost superimposed. The value $\Pi = 0.2$ of Coles' parameter is the same as that proposed from open-channel flow experiments [18]. This provides an explanation about the accuracy of the computed velocity profiles obtained by the first model (Equation (21)) in the outer part of free surface flows.

The profiles obtained for the four $Re_*$. numbers seem similar. For large values of $Re_*$, the two coefficients $C_\alpha = \alpha_1 \kappa$ and $C_1$. of the first model reach asymptotic values

equal, respectively, to $C_\alpha = \alpha_1 \kappa = e^{-(0.34/0.46)} = 0.477$ and $C_1 = 1/0.46 = 2.17$ (Figure 6). Equations (20) and (21) reverts to the following $Re_*$-independent forms:

$$\frac{\nu_t}{h\,u_*} = C_\alpha \xi e^{-C_1 \xi} \tag{23}$$

$$\frac{\nu_t}{h\,u_*} = C_\alpha \xi \left(1 - e^{-B_f(1-\xi)}\right) e^{-C_1 \xi} \tag{24}$$

where $C_\alpha = 0.477$ , $C_1 = 2.17$, and $B_f = 6$.

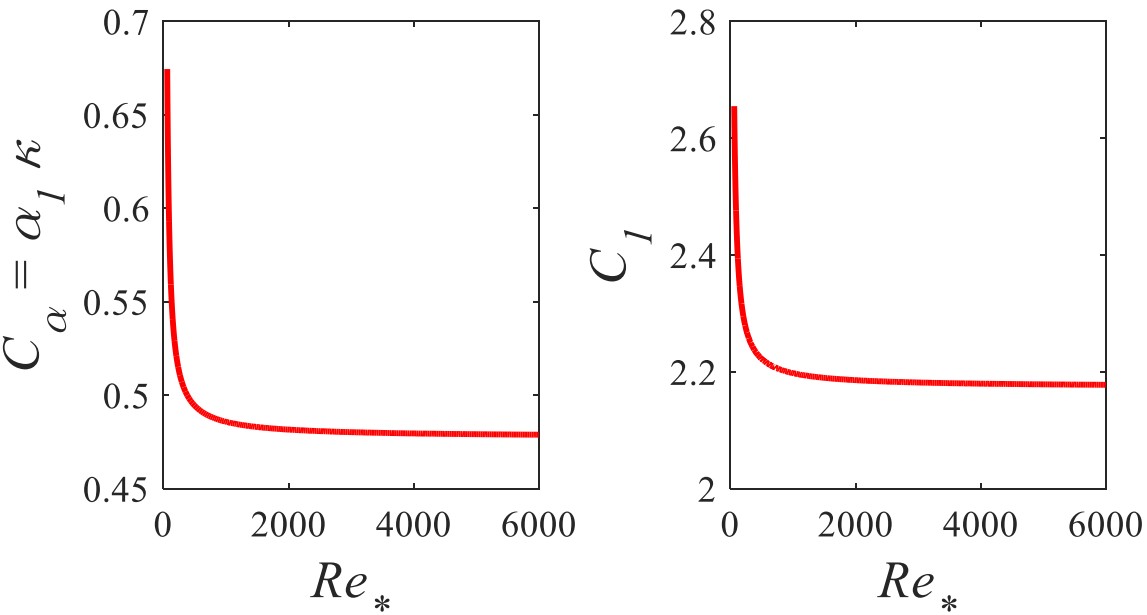

**Figure 6.** Asymptotic behavior of the two coefficients of Equations (23) and (24): $C_\alpha = \alpha_1 \kappa = 0.477$ and $C_1 = 2.17$.

Figure 7 shows the comparison between the two proposed eddy viscosity models (24, 22), both with free surface damping function and parabolic and log-wake-modified eddy viscosity profiles. Figure 7 shows that with the free surface damping function both exhibit smaller values than the parabolic profile. Both models predict profiles similar to log-wake-modified eddy viscosity (Equation (19)). The first model (Equation (24)) (red solid line) is similar to (Equation (19))with a Coles' parameter $\Pi = 0.2$ (red dash-dotted line), while the second model (Equation (22)) (blue solid line) is similar to (Equation (19))with a Coles' parameter $\Pi = 0.15$ (blue dash-dotted line). Interestingly, the two coefficients of the first model are found to be equal, respectively, to $C_\alpha = \alpha_1 \kappa = 0.477$ and $C_1 = 2.17$. The value of the first coefficient $C_\alpha$ results in $\alpha_1 = 1.16$ (with $\kappa = 0.41$), which is close to $\alpha_1 = 1$ (related to the assumption $\sqrt{D_k} = 1/C_\mu^{\frac{1}{4}}$) (see also [49,50]). The value of $C_\alpha$ is also between the two values $0.41 < 0.478 < 0.49$ obtained, respectively, with the assumption $\sqrt{D_k} = 1/C_\mu^{\frac{1}{4}}$, $\sqrt{D_k}C_\mu^{\frac{1}{4}}\kappa = \kappa = 0.41$ and the empirical value $D_k = 4.78$ [56], $\sqrt{D_k}C_\mu^{\frac{1}{4}}\kappa = 0.49$. The value of the second coefficient allows the coefficient in TKE to be defined as $C_k = C_1/2 = 1.088$, which is close to the empirical value $C_k = 1$ [55].

Figure 8 shows a comparison between the two proposed eddy viscosity models (24, 22) and experimental data of eddy viscosity [18,57]. In addition to the experimental data of Nezu and Rodi [18], data from experiments of Ueda et al. [57] are used, which seem to confirm the same behavior. Figure 8 shows that profiles obtained from both models show good agreement with experimental data of eddy viscosity.

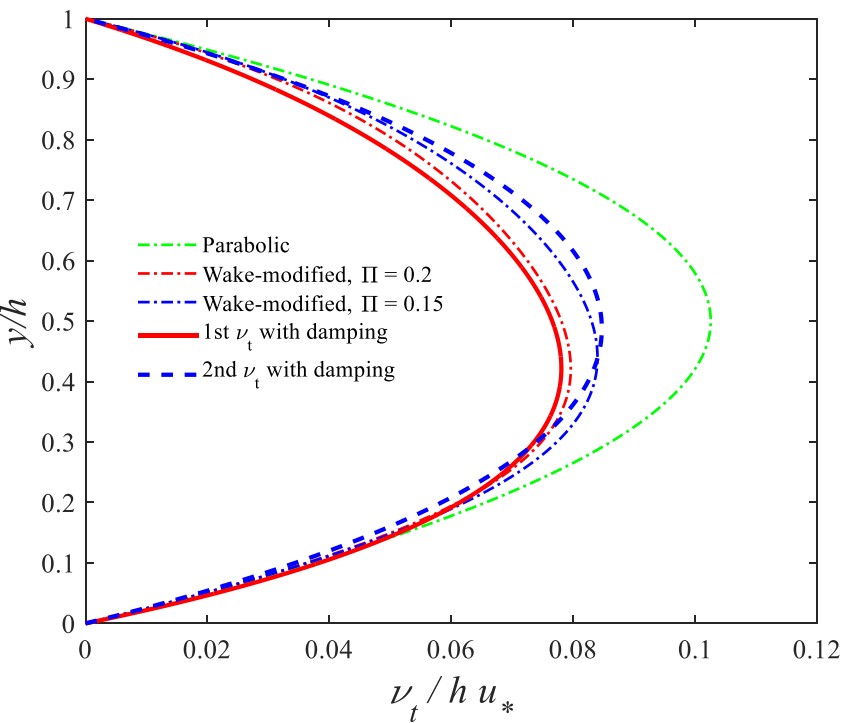

**Figure 7.** Eddy viscosity profiles; red solid line: first eddy viscosity (Equation (24)); blue dashed lines: second eddy viscosity (Equation (22)) both with free surface damping function; red dash-dotted line: log-wake-modified (Equation (2b)) with Π = 0.2; blue dash-dotted line: Equation (2) with Π = 0.15; green dash-dotted line: parabolic eddy viscosity (Equation (18)).

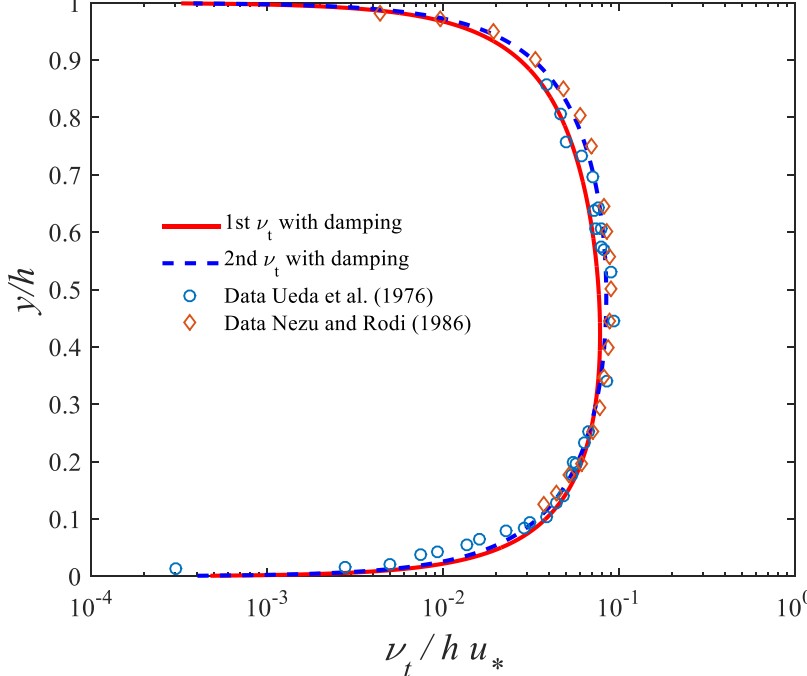

**Figure 8.** Validation by experimental data; eddy viscosity profiles; red solid line: first eddy viscosity (Equation (24)); blue dashed lines: second eddy viscosity (Equation (22)) both with free surface damping function; symbols: experimental data of Nezu and Rodi (1986) and Ueda et al. (1976).

## 5. Conclusions

The parabolic eddy viscosity is based on the log-law velocity profile and is valid only in the log layer. The improved log-wake-modified eddy viscosity was obtained from the log-wake law velocity profile. However, both were obtained from velocity profiles. The main shortcoming of the log-wake profile is the uncertainty in the value of the empirical Coles' parameter.

In this study, the eddy viscosity is defined as a product between a velocity scale (related to the root-square of TKE, which is given by a semi-theoretical exponentially decreasing function) and a length scale (related to mixing length). From this definition, two analytical eddy viscosity models are proposed, namely, (1) the exponential-type profile of eddy viscosity and (2) an eddy viscosity based on an extension of von Karman's similarity hypothesis with an additional correction in order to account for the damping effect of turbulence near the free surface. As for the parabolic and log-wake-modified profiles, the eddy viscosity from the second model is $Re_*$ independent, while the eddy viscosity from the first model is $Re_*$ dependent. The proposed analytical models are validated through computation of velocity profiles, obtained from the resolution of the momentum equation and comparisons to experimental data.

Mean streamwise velocity profiles were obtained by solving the momentum equation. For both the first and second proposed analytical eddy viscosity models, a boundary condition is applied at the lower limit of the outer region, i.e., $y^+ = 0.2\ Re_*$ or $\xi = 0.2$, where the velocity is given by the logarithmic law. Computed mean velocities are compared to experimental data of open-channel flows for $923 < Re_* < 6139$. Computed velocity profiles show good agreement in the outer region.

In order to improve the results, the first eddy viscosity was used with the free surface damping function. Results allow the prediction to be improved, particularly for high Reynolds numbers. For the second analytical eddy viscosity model, a second boundary condition was used at the lower limit of the logarithmic layer, i.e., $y^+ = 30$ where the velocity is given by the log law. Results allow predicted velocity profiles to be improved, particularly for high Reynolds numbers. The results for velocity profiles show the ability of these analytical eddy viscosity models to predict accurately the velocities in the outer region from the momentum equation.

Finally, the vertical distribution of eddy viscosity from both proposed analytical models was analyzed. Both profiles from the first and second analytical eddy viscosity models with the free surface damping function are similar to the log-wake-modified profiles but with different values of Coles' parameter. $\Pi = 0.2$ for the first model and $\Pi = 0.15$ for the second model. These values are close to values found from open-channel flow experiments. This provides an explanation of the accuracy of the computed velocity profiles in the outer part of free surface flows.

For large values of $Re_*$ ($Re_* > 2000$), the first model becomes $Re_*$ independent, and the two coefficients reach asymptotic values equal to $C_\alpha = \alpha_1 \kappa = 0.477$ and $C_1 = 2.17$. Interestingly, with a value of Coles' parameter $\Pi = 0.2$, eddy viscosities from both the first model and log-wake-modified profile are almost superimposed. The value $\Pi = 0.2$ of Coles' parameter is the same as that proposed from open-channel flow experiments. The analysis of these two coefficients allowed the models' assumptions to be verified and former empirical values to be found. The two proposed eddy viscosity models are validated by two experimental data. The comparison shows that profiles from both models show good agreement with experimental data of eddy viscosity.

**Funding:** This research received no external funding.

**Institutional Review Board Statement:** Not applicable.

**Informed Consent Statement:** Not applicable.

**Data Availability Statement:** Not applicable.

**Acknowledgments:** The author would like to thank Xiaofeng Liu from Penn State University for providing the experimental data of eddy viscosity of references [18,57] used in Figure 8.

**Conflicts of Interest:** The author declares no conflict of interest.

**Appendix A**

The von Karman's similarity hypothesis allows writing the mixing length as [58]

$$l_m{}^+ = -\kappa \frac{dU^+/dy^+}{d^2U^+/dy^{+2}} \tag{A1}$$

where $U^+$ is the streamwise mean velocity. With $dU^+/dy^+ \approx \sqrt{k^+}/l_m{}^+$, Equation (A1) becomes

$$l_m{}^+ = -\kappa \frac{\sqrt{k^+}/l_m{}^+}{d\left(\sqrt{k^+}/l_m{}^+\right)/dy^+} \tag{A2}$$

Introducing the function $f^+ = \sqrt{k^+}/l_m{}^+$, (A2) becomes

$$\sqrt{k^+} = -\kappa \frac{f^{+2}}{df^+/dy^+} \tag{A3}$$

We write (A3) in the following form:

$$-\frac{df^+/dy^+}{f^{+2}} = \frac{\kappa}{\sqrt{k^+}} \tag{A4}$$

The integration of the LHS term of Equation (A4) from $A_0$ to $y^+$, is given by

$$\int_{A_0}^{y^+} -\frac{df^+/dy^+}{f^{+2}} dy^+ = \frac{1}{f^+(y^+)} - \frac{1}{f^+(A_0)} \tag{A5}$$

Integrating (A4) from $A_0$ to $y^+$, provides, therefore, the mixing length as

$$l_m{}^+(y^+) = \sqrt{k^+} \left( \kappa \int_{A_0}^{y^+} \frac{1}{\sqrt{k^+}} dy^+ + \frac{l_m{}^+(A_0)}{\sqrt{k^+(A_0)}} \right) \tag{A6}$$

Using Equation (6) for TKE and taking the boundary condition $l_m{}^+(A_0) = \kappa A_0$ yields

$$l_m{}^+(y^+) = \kappa e^{-C_k y^+/Re_*} \left( \frac{Re_*}{C_k} \left( e^{C_k y^+/Re_*} - e^{C_k A_0/Re_*} \right) + A_0 e^{C_k A_0/Re_*} \right) \tag{A7}$$

Rearranging the terms of (A7) allows writing the mixing length as [37–39,54]

$$l_m{}^+(y^+) = \alpha \left( Re_* - (Re_* - C_k A_0)e^{-C_k(y^+ - A_0)/Re_*} \right) \tag{A8}$$

where $\alpha = \kappa/C_k$. Since $A_0 \ll Re_*$ we write (18) in a simplified form that does not depend on $A_0$, as

$$l_m{}^+(y^+) = \alpha \, Re_* \left( 1 - e^{-C_k y^+/Re_*} \right) \tag{A9}$$

By taking $C_k = 1$, Equation (A9) reverts to Equation (13).

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
