# Peer review of "Reinvestigating the Parabolic-Shaped Eddy Viscosity Profile for Free Surface Flows"

_hydrology, doi:10.3390/hydrology8030126_

Round 1

Reviewer 1 Report

The manuscript presents two improved approaches for the estimation of eddy viscosity. It could be relevant for "Hydrology" readership. I think that the manuscript could be considered for publication after minor revisions (see my comments in the attached file).

Author Response

Dear Reviewer 1

We were pleased to know, by a letter from the Editor of Hydrology MDPI, that our manuscript was rated as potentially acceptable for publication in Hydrology MDPI, subject to adequate revision and response to your comments.

Please, find enclosed the revised manuscript with corrections based on your comments.

As you notice, your comments were very useful and identified areas of manuscript that needed clarification. We would like to take this opportunity to express you our sincere thanks.

We hope that the revised manuscript is accepted for publication in Hydrology MDPI.

Sincerely Yours,

R. Absi

Reviewer 2 Report

The author presented a study verifying the effect of modified eddy viscosity by deriving a velocity profile related to eddy viscosity. However, it is difficult to understand this study because sufficient explanations about derivations and assumptions to formulate the eddy viscosity were not introduced even though they are key idea of this study. Also, there are some unclear explanations to follow the research results. The detailed comments are given below.

Line 69: The meaning of the dip-phenomenon is not clear at here. According to Chapter 4, it seems to imply a decrease in velocity near the water surface, but it requires further explanation.

Ch. 2 and 3 are divided into sub-chapters, but each chapter doesn’t provide enough explanations. Ch. 2.2 and 2.3 seem to explain the repeated content. Since providing a vertical distribution of eddy viscosity is an important part of this study, sufficient explanations of the differences between the existing eddy viscosity model and the proposed model in this study and the used assumptions are required.

In title of Ch. 3.3, what is the VKSH?

In Ch. 4, the authors computed stremwise velocity profiles using the proposed eddy viscosity models based on Eq. (16). How to reproduce the velocity profiles? Did the authors solve Eq. (16) in analytically or numerically? If it was solved in analytical method, it would be better to suggest velocity profile formulas.

In Line 172, what is the meaning of the resolution of the momentum equation?

In Table 1, the case number presented from P1 to P4. But, in Figures 1-3, case number shows from P2 to P5.

In Figure 2, the velocity profiles using the exponential type eddy viscosity show velocity reduction near water surface. Is it general property of the exponential type model? Is the damping function required to correct the property? What is the value of damping coefficient used in this study to correct the velocity profile? The velocity profiles shown in Figure 3 based on the second type eddy viscosity model seem to show better results near water surface. Why does the difference between two models occur?

Lines 206-207: Did you mean Figure 3?

A legend should be added in Figure 5.

Figure 7 should be renumbered to Figure 6.

Lines 242-243, Lines 261-262: Any experimental results were not co-plotted with the computed values in Figures 4 and 5. How can the author insist that the computed results properly reproduce the measurements?

Author Response

Dear Reviewer 2

We were pleased to know, by a letter from the Editor of Hydrology MDPI, that our manuscript was rated as potentially acceptable for publication in Hydrology MDPI, subject to adequate revision and response to your comments.

Please, find enclosed the revised manuscript with corrections based on your comments.

Please find the response to your comments and suggestions

The author presented a study verifying the effect of modified eddy viscosity by deriving a velocity profile related to eddy viscosity. However, it is difficult to understand this study because sufficient explanations about derivations and assumptions to formulate the eddy viscosity were not introduced even though they are key idea of this study. Also, there are some unclear explanations to follow the research results. The detailed comments are given below.

Thank you for this comment, additional explanations are introduced in the revised manuscript for the second eddy viscosity in appendix A

Line 69: The meaning of the dip-phenomenon is not clear at here. According to Chapter 4, it seems to imply a decrease in velocity near the water surface, but it requires further explanation.

In the introduction (line 69) we presented the different values of the Coles parameter used in literature. Among these values we found in 2011 that dip-phenomenon requires a larger value equal to 0.45 (please see Absi 2011 Journal of Hydraulic Research).

In section 4 of the manuscript, we presented the general ordinary differential equation (15) with the parameter α related to dip-phenomenon. However, we used equation (16) since the data used in this manuscript are for wide open-channel and therefore α=0.

Ch. 2 and 3 are divided into sub-chapters, but each chapter doesn’t provide enough explanations. Ch. 2.2 and 2.3 seem to explain the repeated content. Since providing a vertical distribution of eddy viscosity is an important part of this study, sufficient explanations of the differences between the existing eddy viscosity model and the proposed model in this study and the used assumptions are required.

Section 2.2 is related to Log-wake modified eddy viscosity profile

Both parabolic eddy viscosity (section 2.1) and Log-wake modified eddy viscosity (section 2.2) are obtained from velocity profiles, respectively log-law and log-wake profiles.

The aim of section 2.3, is to define the eddy viscosity from the concepts of velocity and length scales which are here given by mixing velocity and mixing length. From this definition, we found these two parameters from former eddy viscosity models

The main assumptions in this study are about the shape of mixing velocity and mixing length. The mixing velocity is related to turbulent kinetic energy TKE (equation 7). The mixing length is based on the local equilibrium assumption (1st model) and von Karman similarity hypothesis (2nd model) introduced in appendix A.

In title of Ch. 3.3, what is the VKSH?

Thank you for this comment, it was changed to “von Karman’s similarity hypothesis” in the revised manuscript

In Ch. 4, the authors computed stremwise velocity profiles using the proposed eddy viscosity models based on Eq. (16). How to reproduce the velocity profiles? Did the authors solve Eq. (16) in analytically or numerically? If it was solved in analytical method, it would be better to suggest velocity profile formulas.

Eq. (16) was solved numerically. We added this information in the revised manuscript

In Line 172, what is the meaning of the resolution of the momentum equation?

We added “numerical”:  Mean Streamwise velocities are obtained from numerical resolution of equation (16).

In Table 1, the case number presented from P1 to P4. But, in Figures 1-3, case number shows from P2 to P5.

Thank you for this comment, the mistake is corrected in the revised manuscript, in table 1 case numbers are P2 to P5 and not P1 to P4.

In Figure 2, the velocity profiles using the exponential type eddy viscosity show velocity reduction near water surface. Is it general property of the exponential type model? Is the damping function required to correct the property? What is the value of damping coefficient used in this study to correct the velocity profile? The velocity profiles shown in Figure 3 based on the second type eddy viscosity model seem to show better results near water surface. Why does the difference between two models occur?

In Figure 2, Velocity profiles obtained from (16) with the 1st eddy viscosity (12) (red solid lines) and from (16) with (17) magenta thick dashed lines. The value of damping coefficient is given in the figure caption . With damping function, both models show good results near water surface.

Lines 206-207: Did you mean Figure 3?

Yes thank you, it was corrected, sorry for the mistake

A legend should be added in Figure 5.

Figure 5 contains a legend

Figure 7 should be renumbered to Figure 6.

The first manuscript contains 7 figures. The revised manuscript contains 8 figures.

Lines 242-243, Lines 261-262: Any experimental results were not co-plotted with the computed values in Figures 4 and 5. How can the author insist that the computed results properly reproduce the measurements?

Thank you for this important comment, experimental data of eddy viscosity [18,57] were added in the revised manuscript. In addition to the experimental data of Nezu and Rodi [18], we used data from experiments of Ueda et al. [57] in figure 8.

As you notice, your comments were very useful and identified areas of manuscript that needed clarification. We would like to take this opportunity to express you our sincere thanks.

We hope that the revised manuscript is accepted for publication in Hydrology MDPI.

Sincerely Yours,

  1. Absi

Author Response

Dear Reviewer 3

We were pleased to know, by a letter from the Editor of Hydrology MDPI, that our manuscript was rated as potentially acceptable for publication in Hydrology MDPI, subject to adequate revision and response to your comments.

Please, find enclosed the revised manuscript with corrections based on your comments.

Please find the response to your comments and suggestions

[R1] Results. (i) Test conditions for the 4 experimental runs considered in this study are briefly presented in Table 1. However, the Authors don’t argue at all the nature (measurements included) of these experiments and the negligibility of potential scale effects.

Thank you for this important comment, we added in the revised manuscript more details about the experimental data used in the study.

Additional experimental data of eddy viscosity were added in the revised manuscript (figure 8).

The definition of Scale Effects is given by “the inability to keep each relevant force ratio constant between the scale model and its real-world prototype” (Heller 2011), or the more general definition “the unavoidable difference between model and prototype due to the reduction in size” (Echávez 2012).

However, scale effects are not responsible for discrepancies between physical and numerical model results. These discrepancies “are not scale effects but rather are due to an improper or incomplete numerical, or  theoretical,  model” (Echávez 2012).

Heller V. (2011) “Scale effects in physical hydraulic engineering models”, Journal of Hydraulic Research, 49(3), 293-306.

Echávez G. (2012) “Discsussion of Scale effects in physical hydraulic engineering models By Valentin Heller”, Journal of Hydraulic Research, 50(2), 249-250.

[R2] Table 1. (i) It reads “Froud number”, but it should read “Froude number”!! (ii) The definition of the Froude number is wrong!! The kinematic viscosity there shouldn’t be!

Yes thank you, it was corrected, sorry for the mistake

[R3] Figure 1. Where test conditions for run P5 are specified?

[R4] Figures 2 and 3. As just above, where test conditions for run P5 are specified? Why did P1 and P4 runs are not considered?

Thank you for this comment, the mistake is corrected in the revised manuscript, in table 1 case numbers are P2 to P5 and not P1 to P4.

[r1] Keywords. The keywords “turbulence” and “velocities” appear too generic and ineffective in identifying the crucial issues of this manuscript. I would substitute them for more suitable ones.

Thank you for this comment, “turbulence” is removed and “velocities” is replaced by “streamwise velocity distribution”

As you notice, your comments were very useful and identified areas of manuscript that needed clarification. We would like to take this opportunity to express you our sincere thanks.

We hope that the revised manuscript is accepted for publication in Hydrology MDPI.

Sincerely Yours,

  1. Absi

Round 2

Reviewer 2 Report

The authors adequately revised the manuscript according to the reviewer's comments. It is considered to be possible for publication in present form.

Author Response

Dear Reviewer

We were pleased to know, by a letter from the Editor of Hydrology MDPI, that our manuscript was rated as potentially acceptable for publication in Hydrology MDPI, subject to adequate revision.

Please, find enclosed the revised manuscript with all required minor corrections.

As you notice, your comments were very useful and identified areas of manuscript that needed clarification. We would like to take this opportunity to express you our sincere thanks.

We hope that the revised manuscript is accepted for publication in Hydrology MDPI.

Sincerely Yours,

R. Absi

Reviewer 3 Report

Overall, the Author addressed all my concerns satisfactorily and I find the manuscript is improved. As remarked in my previous review, the MS might be of interest - but perhaps only to the limit - to the Hydrology journal readership. The topic of this study is (clearly) closer to the fluid mechanics than to the hydrology. I have also noted that there are several self-citations. I have even counted 16 self-citations!! Are all these citations strictly necessary?

Nevertheless, I recommend accepting this manuscript nearly in present form. However, some refinements are needed. As minor revisions, I would suggest the following:

At line 62 and all over the MS. It reads “the Coles parameter”, but I would write “the Coles’ parameter” as - for instance – done at line 81.

At line 80. It reads “Our study aims to provide”, but I would like to point out that there is only one Author for this manuscript! Therefore, I would write “This study aims to provide”.

At line 144. It reads “3.4. First formulation”, but it should read “3.3. First formulation”!

At line 160. Similarly, it reads “3.3. Second formulation”, but it should read “3.4. Second formulation”!

At line 161. Is “Absi (2003)” in the References list?

At lines 258 and 259. It reads “1nd”, but it should read “1st”.

At lines 286. Again, it reads “1nd”, but it should read “1st”.

Appendix A. As remarked above, there is only one Author for this manuscript. The use of “we” (e.g. at lines 372 and 377) is therefore improper!

Reference #42. This reference is incomplete! 

Author Response

(The authors gave the same response as above.)
